# Traditional East Asian Herbal Medicine for Post-Stroke Insomnia: A Systematic Review and Meta-Analysis of Randomized Controlled Trials

**DOI:** 10.3390/ijerph19031754

**Published:** 2022-02-03

**Authors:** Sang-Ho Kim, Jung-Hwa Lim

**Affiliations:** 1Department of Neuropsychiatry of Korean Medicine, Pohang Korean Medicine Hospital, Daegu Haany University, 411 Saecheonnyeon-daero, Pohang-si 37685, Korea; omed22@naver.com; 2Department of Neuropsychiatry, School of Korean Medicine, Pusan National University, 49 Busandaehak-ro, Mulgeum-eup, Yangsan-si 50612, Korea

**Keywords:** herbal medicine, stroke, insomnia, systematic review, meta-analysis

## Abstract

Post-stroke insomnia (PSI) is a highly prevalent complication after stroke. Current evidence of psychotropic drug use for PSI management is scarce and indicates harmful adverse events (AEs). Traditional East Asian herbal medicine is a widely used traditional remedy for insomnia. However, so far, no study has systematically reviewed the efficacy and safety of traditional east asian herbal medicine (HM) for PSI. Therefore, we perform meta-analysis to evaluate the effectiveness and safety of HM for PSI. After a comprehensive electronic search of 15 databases, we review the randomized controlled trials (RCTs) of HM use as monotherapy for PSI. Our outcomes were the Pittsburgh sleep quality index and total effective rate. In total, 24 RCTs were conducted with 1942 participants. HM showed statistically significant benefits in sleep quality. It also appeared to be safer than psychotropic drugs in terms of AEs, except when the treatment period was two weeks. The methods used for RCTs were poor, and the quality of evidence assessed was graded “low” or “moderate.” The findings of this review indicate that the use of HM as a monotherapy may have potential benefits in PSI treatment when administered as an alternative to conventional medications. However, considering the methodological quality of the included RCTs, we were uncertain of the clinical evidence. Further, well-designed RCTs are required to confirm these findings.

## 1. Introduction

Stroke is the second-largest cause of death around the world and has a high mortality rate [1,2]. Stroke survivors are commonly affected by neuropsychiatric disorders, with around one-third of them suffering from mental health problems, such as insomnia, depression, or anxiety [3,4,5]. Furthermore, post-stroke insomnia (PSI) is a particularly common complaint of stroke survivors. Insomnia is defined as difficulty in initiating or maintaining sleep, early morning awakening, or suffering from poor quality of sleep for minimum three nights a week over a period of three months such that it affects daytime functioning, according to the standard diagnostic criteria of Diagnostic and Statistical Manual of Mental Disorders [6]. Studies using diagnostic tools estimated the prevalence of PSI as 32.21%. This increased to 40.70% when only insomnia symptoms were considered in a recent meta-analysis [3]. Another meta-analysis found almost similar prevalence rates of insomnia for stroke survivors in the acute, subacute, and chronic phases (40.7%, 42.6%, and 35.9%, respectively) [7]. This means that the sleep difficulties following a stroke may sustain for a long time.

PSI can have negative impact on the recovery, outcome, quality of life (QoL), and even survival of patients. In an animal stroke model, insomnia was associated with brain injury and impaired neuroplasticity [8]. Sleep disturbance affected the improvement of balance and gait in stroke inpatients [9]. Sleep difficulty showed negative impacts on global functional recovery, as measured by the Stroke Impact Scale, in the first year after stroke [10]. Chronic insomnia was associated with increased depression and anxiety, and the inability to return to work even one year after stroke, compared to those with no insomnia [11]. Insomnia significantly lowered the QoL and increased the mortality of stroke patients [12,13,14]. Moreover, insomnia is associated with a post-stroke suicidal tendency [15]. Ischemic stroke is known to disrupt the sleep architecture and endogenous circadian rhythm of patients, suggesting that these conditions may, probably, have a bidirectional relation [16]. Thus, these conditions might have a key role in addressing the PSI of patients in neurorehabilitation following stroke, as well as inpatient management.

Although psychotropic drugs are usually used for the management of PSI, we have scare evidence of their effect, except for one study on the use of selective serotonin reuptake inhibitor (SSRI) drugs [17]. Moreover, we have only pre-clinical evidence of using Zolpidem [18,19]. This drug might neutralize the sleep-dependent neuroplasticity in an animal that suffered a chronic stroke [20] and even increase the risk of ischemic stroke, especially with higher dosages [21]. Benzodiazepines, another commonly used hypnotic drug, have many side effects, such as dependency and rebound symptoms after discontinuation [22]. Zolpidem, zopiclone, and zaleplon are non-benzodiazepine drugs used in the treatment of insomnia and commonly referred to as the “Z-drugs because the generic names of two of the three currently approved agents in the U.S. begin with the letter Z. Although, Z-drugs are sold as alternatives to benzodiazepines, they have been frequently associated with both withdrawal issues and misuse/abuse [23]. Cognitive behavioral therapy for insomnia (CBT-I) is recommended as the primary nonpharmacological treatment for chronic insomnia [24]. However, CBT-I referrals and implementation face several barriers, such as lack of trained practitioners, knowledge, treatment beliefs, motivation, and time [25]. Despite the recent development and delivery of CBT for chronic PSI, relevant evidence is scarce [26]. Therefore, we need a more effective, safe, and convenient alternate therapy for the management of PSI.

Traditional east asian herbal medicine (HM) has been commonly used for the treatment of insomnia for thousands of years in East Asia. It is one of the most widely used forms of complementary and alternative medicine (CAM) [27]. Numerous clinical practice guidelines for insomnia provide CAM recommendations, including HM, for insomnia [28]. As HM works through multi-component and multi-pathway mechanisms, it has been attracting attention as a new alternative to conventional pharmacology [29]. Recent systematic reviews have provided evidence that HM provides significantly better results in insomnia treatment than placebo, with few adverse effects [30]. One retrospective study showed that the use of HM, along with alprazolam, for the treatment of insomnia improves the sleep quality and self-care ability of patients with PSI [31]. 

A previous Chinese study systematically reviewed the use of HM for the treatment of PSI, but this review is outdated, not adequately comprehensive, and pre-registered [32]. Therefore, we need an updated search and comprehensive synthesis to offer optimal recommendations with regard to the efficacy and safety of HM for the treatment of PSI. Our review performs an updated meta-analysis, to provide trustworthy evidence on the effectiveness and safety of HM for the treatment of PSI. 

## 2. Materials and Methods

### 2.1. Study Registration

The protocol for this review was registered with the Open Science Framework (OSF) registries (URL: https://osf.io/8t52f, accessed on 30 January 2022) and International Prospective Register of Systematic Reviews (URL: https://www.crd.york.ac.uk/prospero/display_record.php?ID=CRD42020177862, accessed on 30 January 2022) before commencement of the study. The study protocol was also published [33]. This review is reported in accordance with the Preferred Reporting Items for Systematic Reviews and Meta-Analyses (PRISMA) guidelines (Appendix A) [34]. 

### 2.2. Inclusion and Exclusion Criteria

#### 2.2.1. Types of Studies

We included randomized controlled trials (RCTs) using HM for PSI treatment in this review, regardless of publication or language restrictions. Additionally, we included RCTs only if the expression “randomization” was mentioned without a detailed randomization method. Conversely, we excluded uncontrolled trials, case series, experimental studies, and reviews, as well as quasi-RCTs that use alternate allocation as random sequence generation methods.

#### 2.2.2. Types of Participants

We included patients who were diagnosed with insomnia after stroke using standard imaging tools, such as brain magnetic resonance imaging, brain magnetic resonance angiography, brain computed tomography, or perfusion imaging. We considered only studies using the standardized diagnostic criteria for insomnia, such as the Diagnostic and Statistical Manual of Mental Disorders, International Classification of Diseases, International Classification of Sleep Disorders, Chinese Classification of Mental Disorders, Guideline for the Diagnosis and Treatment of Insomnia in Chinese Adults, or the guiding principles for clinical research on new drugs of traditional Chinese medicine. We excluded all patients with other psychiatric problems, such as depression, anxiety, or other serious illnesses, such as liver disease, kidney disease, or cancer, or drug allergies. We also excluded the patients considered in studies that did not use diagnostic criteria or validated tools. This review has no restrictions regarding age, gender, nor race of participants.

#### 2.2.3. Types of Interventions

##### Experimental Intervention

Studies involving HM as monotherapy, with or without routine care for stroke, such as anti-hypertensives and rehabilitation as interventions, were included. Any HM formulation based on East Asian traditional medicine (EATM) was allowed. Studies involving oral HM along with psychotropic drugs, such as hypnotics, anxiolytics, and antidepressants, and those involving acupuncture/acupressure, moxibustion, herbal bathing, music therapy, and psychotherapy as adjunctive interventions were excluded. Studies that do not list the compositions of the HMs used were excluded, except for patent drugs.

##### Control Intervention

Studies involving psychotropic drugs, such as hypnotics, and antidepressants, anxiolytics, with or without routine care for stroke (RCS), such as anti-hypertensives and rehabilitation, were included.

#### 2.2.4. Types of Outcome Measures

The primary outcome was change in the degree of insomnia as measured using validated assessment tools, such as the Pittsburgh Sleep Quality Index (PSQI) [35], at post-intervention. Total effective rate (TER) and adverse events (AEs) are included as secondary outcome measures. TER is a non-validated outcome measure processed secondarily, based on certain evaluation criteria, such as improvement in quantified outcomes or clinical symptoms. In the assessment of TER, participants are generally classified as “cured,” “markedly improved,” “improved,” or “non-responder” after treatment. TER is calculated consistently using the following formula: TER = N1 + N2 + N3/N, where N1, N2, N3, and N are the number of patients cured, markedly improved, and improved, and the total sample size, respectively. The safety assessment evaluated the incidence of AEs between the experimental intervention and control intervention groups.

### 2.3. Search Methods

One author (SH Kim) searched the following databases from their inception to 9 July 2021: six English databases (Medline via PubMed, EMBASE via Elsevier, the Cochrane Central Register of Controlled Trials (CENTRAL), Allied and Complementary Medicine Database (AMED) via EBSCO, Cumulative Index to Nursing and Allied Health Literature (CINAHL) via EBSCO, and PsycARTICLES via ProQuest), five Korean-databases (Oriental Medicine Advanced Searching Integrated System (OASIS), Korean Medical Database (KMbase), Korean Studies Research Information Service System (RISS), Information Service System (KISS), and Korea Citation Index (KCI)), three Chinese databases (China National Knowledge Infrastructure (CNKI), Wanfang Data, and the Chongqing VIP Chinese Science Database (VIP Database)), and one Japanese database (Citation Information by NII (CiNii)). We also manually searched the reference lists of the eligible studies, relevant systematic reviews, and the ongoing registered clinical trials on the registry to find additional eligible studies. We did not restrict on language or publication status. We used the following search terms in Medline: (“sleep” (MH) OR “sleep wake disorders” (MH) OR insomnia* OR sleep* OR sleepless* OR wakeful* OR dyssomn*) AND (“stroke” (MH) OR stroke) AND (“Plants, Medicinal” (MH) OR “Drugs, Chinese Herbal” (MH) OR “Medicine, Chinese Traditional” (MH) OR “Medicine, Kampo” (MH) OR “Medicine, Korean Traditional” (MH) OR “Herbal Medicine” (MH) OR “Prescription Drugs” (MH) OR “traditional Korean medicine” OR “traditional Chinese medicine” OR “traditional Oriental medicine” OR “Kampo medicine” OR “alternative medicine” OR “complementary medicine” OR herb* OR botanic* OR decoction*) (Appendix A).

### 2.4. Date Collection and Analysis

#### 2.4.1. The Selection of Literature

All retrieved studies were imported into EndNote X20 (Clarivate Analytics, Philadelphia, PA, USA); next, the duplicated studies were excluded using the “Find Duplicates” function in EndNote X20 and manual searching. Two authors (S.H. Kim and J.H. Lim) screened the titles and/or abstracts of studies retrieved independently to identify the studies that potentially meet the inclusion criteria outlined above. They then retrieved the full texts of these potentially eligible studies and independently assessed them for eligibility. Any disagreement between the two authors regarding eligibility of particular studies in the selection process was resolved through consensus.

#### 2.4.2. Data Extraction

We extracted data from the included studies using a pre-defined data collection form in Excel 2016 (Microsoft, Redmond, WA, USA). One author (S.H. Kim) conducted data extraction, and the other (J.H. Lim) reviewed the data. Disagreements were resolved through discussions between them. The items extracted from each study included the first author’s name; year of publication; country; sample size and number of dropouts; diagnostic criteria for insomnia; stroke type; details of participants, experimental intervention, and comparisons; duration of intervention; follow-up; outcome measures; results; AEs associated with the interventions; and information for the assessment of the risk of bias (ROB). We obtained missing data from the study authors via e-mail if additional information was considered necessary. Discrepancies between the authors regarding the process of data extraction were resolved via discussions.

### 2.5. ROB Assessment

The methodological quality was independently assessed using the ROB tool of the Cochrane group by two authors (S.H. Kim and J.H. Lim) [36]. Disagreements were resolved through discussions between them. The ROB tool includes each of the six domains (generation of the allocation sequence, concealment of the allocation sequence, blinding, incomplete outcome data, selective outcome reporting, and other biases). We assessed the study to have a high ROB in the random sequence generation domain when only the expression “randomization” was mentioned, without specifying the randomization methods used. Other potential bias categories were particularly assessed by the emphasis on baseline imbalances between the experimental and control group, such as participant characteristics, including the mean age and baseline insomnia level, based on PSQI, Insomnia Severity Index (ISI), and the Athens Insomnia Scale (AIS).

### 2.6. Data Analysis

#### 2.6.1. Measures of Treatment Effect

We pooled the data with mean difference (MD) for continuous outcomes and risk ratio (RR) for binary outcomes with 95% confidence intervals (CIs).

#### 2.6.2. Assessment of Heterogeneity

The heterogeneity between studies were assessed by using both the chi-squared test and I-squared statistic. We considered I-squared values of ≥50% and ≥75% as indicative of substantial and serious heterogeneity, respectively.

#### 2.6.3. Data Synthesis

First, a narrative synthesis of the findings from the included studies structured around the demographic characteristics of the participants, details of experimental and control interventions, outcomes, and results were provided. In addition, we performed meta-analysis using the Review Manager software version 5.4 for Windows (Copenhagen, The Nordic Cochrane Centre, the Cochrane Collaboration, 2020) when the studies used the same type of experimental intervention, comparison, and outcome measure. We assumed that the true effect size varies from one study to the next, and that the studies in our analysis represent a random sample of effect sizes that could have been observed, because there is significant clinical heterogeneity across included studies. Therefore the data were pooled using a random-effects model, regardless of the heterogeneity. However, the data were pooled using a fixed-effects model when the number of studies included in the meta-analysis was extremely small, meaning that the estimate of the between-study variance lacked precision.

#### 2.6.4. Subgroup Analysis

We conducted a subgroup analysis considering the duration of treatment, the type of psychotropic drug, and the HM prescription. The duration of treatment was classified as 2–3 weeks, 4 weeks, and >8 weeks, respectively, and the psychotropic drugs alprazolam, estazolam, and zopiclone were investigated. We allowed a modified decoction regarding the similarity of HM prescription types.

#### 2.6.5. Sensitivity Analysis

Additionally, we conducted sensitivity analysis by excluding the (1) studies that were rated as high ROB based on selection bias and (2) studies that were numerically distant from the rest data (MD and RR of each study result is −4 points and ≥ 1.5, respectively).

#### 2.6.6. Publication Bias

When more than 10 trials were included in the meta-analysis, we assessed the evidence of publication bias using funnel plots. Additionally, Egger’s linear regression was used to assess the publication bias using R version 4.1.1 software [37].

#### 2.6.7. Summary of Evidence

We used the Grading of Recommendation, Assessment, Development, and Evaluation (GRADE) profiler version 3.6.1 (GRADE Working Group) to assess the quality of evidence [38]. As for assessment scale, the confidence in each outcome was divided into four levels: high, medium, low, and extremely low. The GRADE process begins with asking an explicit question, including specification of all important outcomes. After the evidence is collected and summarized, GRADE provides explicit criteria for rating the quality of evidence that include study design, risk of bias, imprecision, inconsistency, indirectness, and magnitude of effect. The GRADE system rates the quality of evidence for each outcome, from a rating of HIGH to VERY LOW. GRADE starts with a baseline rating of HIGH for RCTs, and LOW for non-RCTs. This baseline rating can then be adjusted (downgraded or, less commonly, upgraded) after considering 8 assessment criteria. Reasons to downgrade the evidence quality are (1) Risk of Bias, (2) Inconsistency, (3) Indirectness, (4) Imprecision, and (5) Publication Bias. In addition, reasons to upgrade the evidence quality are (6) Large Magnitude of Effect, (7) Dose Response, and (8) Effect of all plausible confounding factors would be to reduce the effect (where an effect is observed) or suggest a spurious effect (when no effect is observed). After that, we finally did make a judgement about quality based on these.

## 3. Results

### 3.1. Study Selection

We found 4609 studies through a database search. There were no additional records from other sources. After removing all duplicates and screening the titles and abstracts of 2788 studies, we assessed the full texts of the remaining 98 studies for final inclusion. From these, we again excluded 74 articles for the following reasons: 14 for not involving RCTs; two for not considering PSI; six for considering moderate or higher depression (Hamilton Depression Rating Scale) and anxiety; one for considering cognitive disorder; five for not reporting the diagnostic criteria of stroke; 28 for not reporting the diagnostic criteria of insomnia; three for comparing different HMs; one for using herbal injection as RCS; 13 for using HM and psychotropic drugs in combination; five for using interventions other than HM, psychotropic drugs, and RCS; and one for using duplicate data (Appendix A). Finally, we considered 24 RCTs with 1942 participants in this review (Figure 1) [39,40,41,42,43,44,45,46,47,48,49,50,51,52,53,54,55,56,57,58,59,60,61,62]. 

### 3.2. Study Characteristics

All of the included studies were conducted in China and compared HM with psychotropic drugs. Of these, one was a placebo-controlled 3-arm trial comparing HM, placebo, and the zopiclone group [41]. Three studies [41,43,57] reported that they obtained approval of the institutional review board, and 13 [40,41,42,43,44,47,50,52,53,57,59,60,61] reported that they had obtained consent of the participants. Sample sizes ranged from 60 to 142, with a median of 81. Thirteen studies [40,41,42,44,47,50,52,53,55,56,57,60,61] recruited participants through pattern identification; the most common pattern identification was blood stasis due to Qi or Yin deficiency or liver stagnation in five studies [40,47,55,56,57]; followed by phlegm-heat in four studies [42,44,50,53]; liver stagnation in three studies [44,55,61]; blood deficiency and liver-heat syndrome in one study [41], and heart-kidney non-interaction, in one study [53]. Pattern identification categorizes the symptoms and signs of patients into a chain of syndrome concepts enabling individual treatment [63]. Thirteen studies [40,44,46,49,50,51,52,53,54,55,56,57,60] reported RCS in both HM and control groups. In most cases, RCS used pharmaceutical anti-coagulation, anti-platelet, vasodilators, and neurotrophic agents. As control interventions, three types of benzodiazepines (estazolam, alprazolam, diazepam) and one type of selective gamma-aminobutyric acid receptor (GABA)-A agonist (zopiclone) were used as follows: estazolam in ten studies [39,42,43,44,45,47,48,49,56,61], alprazolam in nine studies [40,46,50,51,54,55,57,58,60], diazepam in two studies [59,62], and zopiclone in three studies [41,52,53]. TER was the most frequently used outcome in 20 studies [39,40,42,43,44,45,46,47,48,49,51,53,54,55,56,57,58,59,60,61], followed by PSQI in 19 studies [39,40,41,42,43,44,47,48,49,50,52,53,54,55,56,57,60,61,62], Insomnia Severity Index (ISI) in two studies [41,43], Athens Insomnia Scale (AIS) in one study [46], polysomnography (PSG) in one study [52], Spigel Sleep Questionnaire in one study [55], Traditional Chinese medicine (TCM) symptom score in five studies [40,52,53,56,60], and Stroke Specific Quality of Life Scale (SS-QOL) in one study [61]. Five different TER calculation methods were used. The TER calculation was based on PSQI in eight studies [42,43,44,47,52,53,57,60], total sleeping time and sleep quality in eleven studies [39,41,42,45,48,50,54,56,58,59,61], total sleeping time and clinical symptoms in one study [55], and AIS in one study [46], while one study [51] did not report its TER calculation method. The treatment duration ranged from 2–12 weeks; the most common period was four weeks (one month) in 18 studies [40,41,44,45,46,47,48,49,50,52,53,54,56,57,58,59,60,61,62], followed by two weeks in three studies [42,43,51]; three weeks in one study [39]; eight weeks in one study [59]; and twelve weeks in one study [55]. After final treatment, a follow-up was performed in three studies [44,61,62], in all of which the duration was four weeks. Twelve studies [39,41,44,45,47,50,52,53,54,56,60,61] reported adverse events (Table 1). Various types of HMs were used in the included studies, of which Chaihu Longgumuli decoction (or pill) was the most frequently used, in three studies [48,51,61]. In terms of dosage form, decoction was most often used, in 19 studies [40,41,42,43,44,45,46,47,48,49,50,53,55,56,57,58,59,61,62], followed by granule in two studies [52,59], pill in two studies [51,54], and capsule in one study [39] (Appendix A). We also analyzed the usage frequency of single HM in HM prescriptions (Appendix A). The most commonly used herbs in the included study were Zizyphi Spinosae Semen, followed by Glycyrrhizae Radix, Rhizoma Salviae Miltiorrhizae, Poria(Hoelen), Acori Graminei Rhizoma, Ligustici Rhizoma, Bupleuri Radix, Polygoni Multiflori Ramuls, and so on. 

### 3.3. ROB in Studies

We evaluated 13 studies [39,41,42,43,46,49,50,52,53,55,57,60,62] using appropriate random sequence generation methods, such as computer-generated random number tables, to find a low ROB in the random sequence generation domain. Six studies [41,45,52,58,62] indicating allocation concealment were evaluated as having low ROB in the allocation concealment domain. All studies were evaluated as having high ROB in blinding of participants because they did not use an herbal placebo. We have evaluated only three studies [42,52,59] as having low ROB in blinded outcome assessors. In two studies reporting dropout [53,60], we rated the domains of incomplete outcome data as low because the missing data were extremely small and similar in both groups. In two other studies that did not report dropout [52,62], the domains of incomplete outcome data were rated as having unclear ROB. No RCTs published their study protocols. We evaluated all studies, except one [41], as having unclear ROB in the selective reporting domain because the protocol of each study was not prospectively registered. One study [51] that did not report its mean age in the intervention and control group was evaluated as having unclear ROB in the domains of the other potential source bias because we could not confirm a significant baseline difference in demographic data between the two groups. We contacted the corresponding authors of one study via e-mail for that information but received no reply (Figure 2).

### 3.4. Effectiveness and Safety of HM

#### 3.4.1. Effectiveness

In meta-analysis, HM showed significant benefits in sleep quality, as assessed by PSQI. This is the primary outcome compared to psychotropic drug group (19 studies [39,40,41,42,43,44,47,48,49,50,52,53,54,55,56,57,60,61,62]; MD −1.90, 95% CI −2.43 to −1.37, I^2^ = 90%), regardless of the treatment duration and type of control medication (Figure 3). The TER in the HM group was also statistically higher (20 studies [39,40,42,43,44,45,46,47,48,49,51,53,54,55,56,57,58,59,60,61]; RR 1.12, 95% CI 1.08 to 1.17, I^2^ = 0%) than the corresponding scores in the psychotropic drug group (Figure 4). Subgroup analysis showed superior effectiveness of HM as demonstrated by PSQI and TER throughout the 12 weeks of treatment. However, in subgroup analysis by type of control medication, the significant between-group differences disappeared in the case of eszopiclone (three studies [41,52,53]; MD −0.46, 95% CI −1.11 to 1.19, I^2^ = 0%). Subgroup analysis according to the type of HM prescriptions showed that the significant between-group differences based on PSQI or TER disappeared in the case of the Wendan decoction (two studies [50,52]; MD −0.73, 95% CI −2.00 to 0.54, I^2^ = 35%) and were maintained in the case of the Chaihu Longgumuli decoction (two studies [48,61]; MD −1.63, 95% CI −2.06 to −0.58, I^2^ =84%/three studies [48,51,61] RR 1.17, 95% CI 1.06 to 1.30, I^2^ = 0%) (Figure 5). 

We confirmed the robustness of these results through sensitivity analyses performed after excluding the low-quality RCTs that were rated as high ROB based on selection bias and studies that were numerically distant from the rest data. Nevertheless, the superior effectiveness of HM demonstrated by PSQI and TER was consistent (Appendix A). Meta-analysis carried out in three follow-up studies (duration of four weeks for all) showed significant benefits of HM in improving sleep quality, as assessed by PSQI, compared to psychotropic drugs in three studies (references [44,61,62]; MD −3.08, 95% CI −3.52 to −2.64, I^2^ = 80%) (Appendix A). Of these three studies, two [61,62] showed significant differences between HM and psychotropic drugs in improving sleep quality; but the other one [44] showed no differences. 

#### 3.4.2. Safety

We found significantly fewer AEs associated with HM (11 studies [39,41,44,45,47,50,52,53,54,60,61]; RR 0.25, 95% CI 0.18 to 0.48) than psychotropic drugs (Figure 6). Subgroup analysis by type of control medication showed superior safety of HM consistently. However, the significant differences between these two groups disappeared when the treatment period was two weeks (one study [39]; RR 0.36, 95% CI 0.10 to 1.25). Subgroup analysis by type of HM prescription, such as Wendan decoction [50,52] and Chaihu Longgumuli decoction [61], did not consistently show superior HM safety. We confirmed the robustness of these results through sensitivity analyses, after excluding low-quality RCTs that were rated as having high ROB based on selection bias. Nevertheless, the superior safety of HM, as demonstrated by fewer AEs, was consistent (Appendix A).

One study assessed safety using Treatment-Emergent Signs and Symptoms (TESS [56]. In this study, HM showed significant safety benefits compared with psychotropic drugs. The AEs associated with HM included upper abdominal or stomach discomfort (5 cases), diarrhea (2 cases), nausea (1 case), and dyspepsia (1 case). The AEs associated with psychotropic drugs included dizziness (13 cases), fatigue (10 cases), sleepiness (10 cases), dry mouth (9 cases), headache (4 cases), and nausea (2 cases). All AEs associated with HM included gastrointestinal symptoms, and the AEs associated with psychotropic drugs included various symptoms, such as neurological, and general symptoms.

### 3.5. Quality of Evidence

In comparisons between HM and psychotropic drug use, the quality of evidence was graded as “low” or “moderate” (Table 2). The quality of evidence regarding outcomes in case of no subgroup analysis was graded as “moderate”. However, the quality of several evidence in the subgroup was graded as low. Thus, we find no high quality of evidence. The main reason for this was the high ROB in the RCTs considered in each meta-analysis. The included study(ies) had a risk of selection and performance bias. Moreover, most findings in subgroup analysis were judged as having low precision because they had wide CIs and did not satisfy the optimal sample size.

### 3.6. Publication Bias

The funnel plot was based on RCTs reporting under PSQI and TER (Figure 7). Since this was generally symmetrical, indicating low risk of publication bias, there were no emerged evidence of publication bias from the funnel plots of PSQI and TER when comparing the efficacy of HM with that of pharmacotherapy. Additionally, publication bias could not be proven using Egger’s method (*p* value for PSQI and TER bias: 0.1656 and 0.4157, respectively) (Table 3). The Egger test results of slop = 0.002, bias (*p*>|t| = 106 > 0.05), 95%CI [−1.53, 0.169], including 0, indicated that there was no obvious publication bias in the literature and confirmed the robustness of our results. However, publication bias can still arise from the publishing factors of these RCTs.

## 4. Discussion

### 4.1. Summary of Evidence

In this systematic review, we performed an extremely comprehensive review and meta-analysis to analyze the effectiveness (or efficacy) and safety of HM in the treatment of PSI. A comprehensive search yielded 24 RCTs suitable for inclusion in our review and meta-analysis. Our meta-analysis indicated that there was a moderate level of evidence that HM intake is associated with a mean reduction of 1.9 points compared with the control medication (MD −1.9 points, 95% CI −1.37 to −2.43), and the TER in HM group was also statistically higher than the corresponding scores in the psychotropic drug group (RR 1.12, 95% CI 1.08 to 1.17). HM showed statistically significant benefits in sleep quality as assessed by PSQI or TER, as well as subgroup analysis of treatment duration and type of control medication, compared to psychotropic drugs. However, in the subgroup analysis of PSQI, a few studies using eszopiclone as control medication did not show statistically significant improvement in patients with PSI as compared to psychotropic drugs. Meta-analysis of a few studies showed significant benefits of HM, as assessed under PSQI, compared to psychotropic drugs, sustaining for four weeks after treatment completion. Additionally, HM appeared safer than psychotropic drugs in terms of AE incidence, except for when the treatment period was two weeks. The methodological quality of RCTs in this systematic review was poor overall because the allocation concealment and selective reporting domains were evaluated as unclear in most studies, and the blinding domains were evaluated as having high ROB. The quality of evidence assessed using GRADE was “Low” or “Moderate”, with no “High” quality evidence. Furthermore, no evidence of publication bias emerged from the funnel plots nor Egger’s method.

### 4.2. Interpretation in Context of Previous Evidences

Our meta-analysis results provide limited evidence of HM as monotherapy being beneficial in PSI treatment compared to psychotropic drugs, especially within the first 2–4 weeks of treatments. This finding is consistent with the findings of previous systematic reviews comparing overall HM with benzodiazepine drugs for primary insomnia or insomnia disorder [64]. In the previous reviews, HM was found to be more effective than benzodiazepine drugs in reducing PSQI scores (MD: −1.94, 95% CI: −2.45 to −1.43). In our meta-analysis, the mean reduction of 1.9 points in PSQI score was below 3 points, indicating a minimal clinically important difference in patients with primary insomnia [65]. Although the TER in the HM group, which may reflect changes in insomnia in the clinical setting, was higher than the TER in the medication group, we were uncertain of this evidence because its quality was not high.

The effect sizes based on the PSQI of each HM prescription for treating insomnia in previous systematic reviews have differed. The effect size of the Chaihu-Longgu-Muli decoction (MD = −2.80, 95% CI −5.48, −0.13, *p* = 0.04) [66] was larger than that of our results, while the results of Xiao-yao-san (MD: −1.82; 95% CI −2.39 to −1.24; *p* < 0.001) [67] was similar to ours. However, those of Shumian capsules (MD = −0.50, 95% confidence interval (CI) = [−0.78, −0.22], *p* = 0.0005) [68] and the Banxia Formulae (MD  =  −1.05, 95% CI −1.63 to −0.47) [69] were smaller than our results. The population of these studies did not include patients with insomnia accompanied by diseases, such as PSI, but, rather, patients with primary insomnia or insomnia disorder were evaluated. Thus, to interpret our primary results, we must consider the heterogeneity and different effect sizes of each HM prescription.

### 4.3. Clinical Implications

HM is a representative EATM method used for many years in Asian countries for the treatment of diseases, such as stroke and insomnia. Our findings suggest that HM may be used for stroke survivors with insomnia as a potential alternative to psychotropic drugs. As insomnia is a potentially modifiable risk factor, we need to address PSI for improvement in stroke symptoms. Current evidence on psychotropic drugs for the management of PSI is scarce [17]. Moreover, psychotropic drugs, such as benzodiazepine and Z-drugs, are involved in harmful adverse events, abuse/dependence risk, and withdrawal symptoms, as well as ischemic stroke, in the case of higher dosage of zolpidem [21,22,23]. Stroke is prevalent in the elderly, with stroke incidence doubling in people aged over 80 [2]. Elderly patients are vulnerable to higher risk of fractures, falls, dementia (at highest doses), pneumonia (in patients with Alzheimer disease), and suicidal tendencies associated with benzodiazepine or Z-drugs [70,71,72,73]. Thus, benzodiazepines and Z-drugs should be prescribed for the elderly with extreme care. Although clinical evidence supporting the effectiveness (or efficacy) and safety of HM for PSI is insufficient, this finding provides clinical evidence that HM may be considered a potentially safe alternative to conventional hypnosedatives for the management of PSI. Additionally, the benefits of HM assessed using PSQI continued for four weeks after completion of treatment. Previous systematic reviews have shown that the effect of HM sustained well in medium- to-long-term follow-up studies [64]. Although the duration of follow-up is short and only three studies were included in our analysis, our findings show that HM may be an effective and safe alternative that has no drug dependency.

### 4.4. The Underlying Mechanism of HM

Numerous preclinical studies have examined the underlying mechanism of various HM’s action: (1) Suanzaoren decoction could induce sleep through anxiolytic-like activities and regulation of neurotransmitter levels, for example, glutamate and γ-aminobutyric acid (GABA) [74]. (2) Suanzaoren decoction regulated the circadian rhythm by regulating the mRNA and protein expressions in the suprachiasmatic nucleus [75]. (3) Chaihu-Longgu-muli decoction inhibits adrenocorticotropic hormone and corticosterone, both of which may be involved in regulating insomnia [76]. (4) Jiaotai Pill alleviated insomnia by regulating the monoaminergic system and activities of organic cation transporters in the hypothalamus and peripheral organs [77]. (5) Jiaotai Pill might promote the repair of intestinal epithelial barriers and suppress the systemic inflammation and cognitive impairment in sleep-deprived rats [78]. 

Sanjoinine A, JuA, and flavonoids in the water extract of zizyphi spinosae semen, the most commonly used herb in our studies, exerted sedative-hypnotic effects by increasing chloride influx and over-expression of α- and γ-subunit GABA receptors in the GABAergic and serotonergic systems [79]. Furthermore, the combination of zizyphi spinosae semen and rhizoma salviae miltiorrhizae showed a significant synergistic effect (*p* < 0.05) in decreasing sleep latency and increasing sleeping time in a mouse model using the pentobarbital-induced sleep method [80]. The sedative-hypnotic effect of HM (herbal pair of semen ziziphi spinosae and radix polygalae) was possibly related to the adjustment of the neurotransmitters 5-hydroxytryptamine, norepinephrine, and dopamine in the total protein of mouse brain tissue [81]. Poria cocos (hoelen) enhances pentobarbital-induced sleeping behaviors via GABAergic mechanisms in rodents [82]. The central inhibitory effects of acori graminei rhizoma were probably affected by an action on the central dopamine and GABA(A) receptors [83]. Saikosaponin a, an active component of bupleuri radix, increased the duration of non-rapid eye movement sleep and shortened sleep latency by decreasing neuronal activity in the lateral hypothalamus in the male C57BL/6j mouse model [84].

HM is, thus, believed to improve sleep through the above diverse and multiple mechanisms.

### 4.5. Strengths and Limitations

Our study is the first pre-registered, updated, and comprehensive systematic review and meta-analysis providing reliable evidence on the effectiveness and safety of HM in treating PSI. Our subgroup analysis and meta-analysis provide information on treatment duration and superiority by type of each control medication. Considering the limitations in use of psychotropic drugs for the management of PSI, particularly in the elderly with stroke, our findings might help provide limited evidence for optimal recommendations of HM as an alternative to benzodiazepines or Z-drugs in clinical practice for the treatment of PSI. 

However, our findings should be interpreted with care because of the following limitations: First, the studies included are generally low in quality, particularly with regard to performance bias due to lack of placebo-controlled trials and selection bias due to improper randomization and allocation. A placebo group was used as a control in only one of the included studies [41], indicating the possibility of our review overestimating the effectiveness of HM. No evidence of “high” quality is observed in the evaluation using GRADE. In other words, our results have low reliability; therefore, they should be interpreted with extreme care because they might change following the findings of future rigorous studies. Second, although no evidence of publication bias is observed, we should be aware of a potential publication bias in the results and in their general application to other countries because all the included studies were conducted in China. Furthermore, the popularity of HM as stroke medicine in China [74] may have elevated the Chinese participants’ expectations of HM and probably increased the placebo effect. In the future, we need to conduct further studies in countries other than China with different rationales. Third, we conducted subgroup and sensitivity analysis, but the composition of HMs used in most studies is heterogeneous. Moreover, about half of the included studies recruited PSI patients with a specific TCM pattern, and the remaining studies recruited PSI patients with no specific TCM pattern. The RCS using HM is also not standardized. These could cause clinical heterogeneity in meta-analyses. Therefore, our results presented the average effects of HM on PSI estimated by a random-effect model, rather than the typical effect. Fourth, as the type of stroke (cerebral infarction or hemorrhage) is mixed in most studies, we did not conduct subgroup analysis according to the type of stroke. We doubt whether HM could be more beneficial to certain types of strokes. Fifth, as there was insufficient information, we did not conduct subgroup analysis according to the severity of insomnia. Sixth, our meta-analysis did not utilize objective measurements, such as polysomnography, but subjective measurements, such as PSQI and TER. Only one of the included studies reported that HM is comparable to zopiclone in all parameters of polysomnography [52]. Therefore, HM may affect not only sleep problems but also neurological disorders. However, as there was insufficient information, we did not conduct meta-analysis for neurological functions in this review. Finally, although 13 studies conducted follow-up ranging from one week to six months after the end of the treatment in a previous review [64], only three studies conducted follow-up with a duration of four weeks. Therefore, we could not discern significant benefits of HM compared to psychotropic drugs that sustained for a long time after treatment completion. 

### 4.6. Implications for Future Research and Practice

Our suggestions for future research are as follows: First, further high-quality RCTs on the efficacy of HM for improving PSI should be conducted, particularly in countries other than China. Double-blind placebo-controlled trials are necessary to assess the efficacy and safety of HM and avoid potential placebo effects. Dai et al.’s study [41], considered in our review, provides a good model to conduct robust RCT. Second, future studies should consider the clinical features of patients and interventions in more detail than was possible in our analysis, particularly regarding the type and stage of stroke, severity of insomnia, specific herbal formulas, and TCM patterns. These findings will provide practical evidence on the optimal use of HM for PSI in clinical settings. The use of standardized HM, in particular, is emphasized to establish an effective HM use strategy for PSI treatment and confirm its expected effectiveness and safety. This review provides clinical information of some candidates for standardized herbal formulas, such as Chaihu-Longgu-muli decoction [66], Ningxin Anshen formula [41], Suanzaoren decoction [85], Jiaotai decoction [86], and insomnia granules. Third, future clinical trials should evaluate the changes in insomnia following the use of objective sleep measurements, such as actigraphy or polysomnography. A study protocol for RCT suggests that HM (Jiao-tai-wan) should explore the efficacy and safety of Jiao-tai-wan in ameliorating insomnia symptoms with the use of PSQI, as well as polysomnography [79]. Fourth, considering the multi-component, multi-target, and multi-pathway characteristics of HM, future studies should evaluate not only the sleep but also neurological disorders, as well as other stroke complications, as outcome. Finally, future studies should conduct long-term follow-up to prove the sustainability of the effectiveness and safety of HM for PSI. A nationwide community-based prospective cohort study of stroke cases would help in addressing such stroke-related issues [74]. 

## 5. Conclusions

The findings of this review indicate that the use of HM as a monotherapy may have potential benefits in PSI treatment when administered as an alternative to conventional medications, such as benzodiazepines and eszopiclone. However, considering the methodological quality of the included RCTs and the fact that no placebo-controlled trials were conducted, we were uncertain of the clinical evidence. Therefore, further well-designed RCTs must confirm the above findings.

## Figures and Tables

**Figure 1 ijerph-19-01754-f001:**
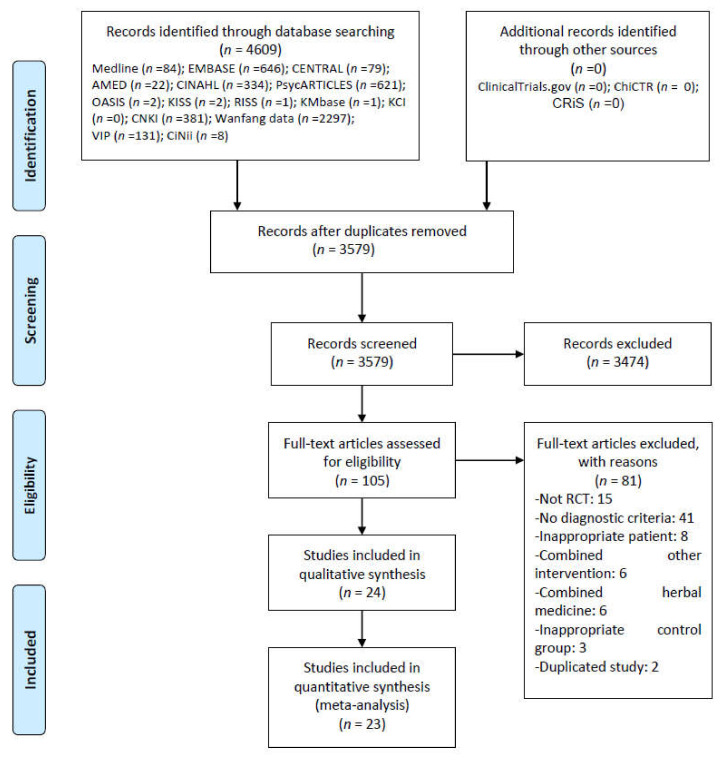
PRISMA flow chart for the study selection process. AMED = Allied and Complementary Medicine Database; CENTRAL = Cochrane Central Register of Controlled Trials; CINAHL = Cumulative Index to Nursing and Allied Health Literature; CiNii = Citation Information by NII; CNKI = China National Knowledge Infrastructure; HM = Herbal Medicine; KCI = Korea Citation Index; KISS = Korean studies Information Service System; KMbase = Korean Medical Database; OASIS = Oriental Medicine Advanced Searching Integrated System; RCT = Randomized Controlled Trial; RISS = Research Information Service System; VIP = the Chongqing VIP Chinese Science Database.

**Figure 2 ijerph-19-01754-f002:**
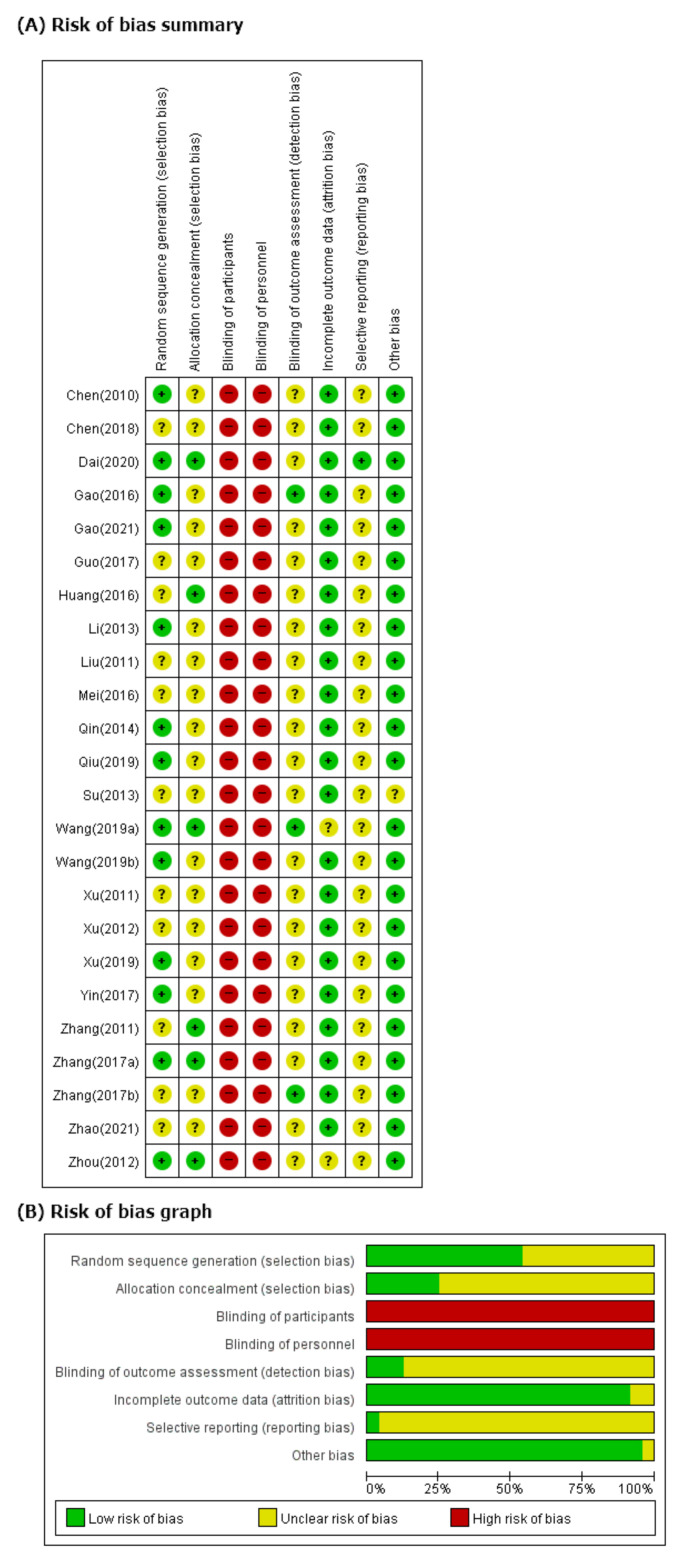
(**A**) Risk of bias summary. Low, unclear, and high risk, respectively, are represented with the following symbols: “+”, “?”, and “−”. (**B**) Risk of bias graph. Review of authors’ judgments about each risk-of-bias item presented as percentages across all included studies.

**Figure 3 ijerph-19-01754-f003:**
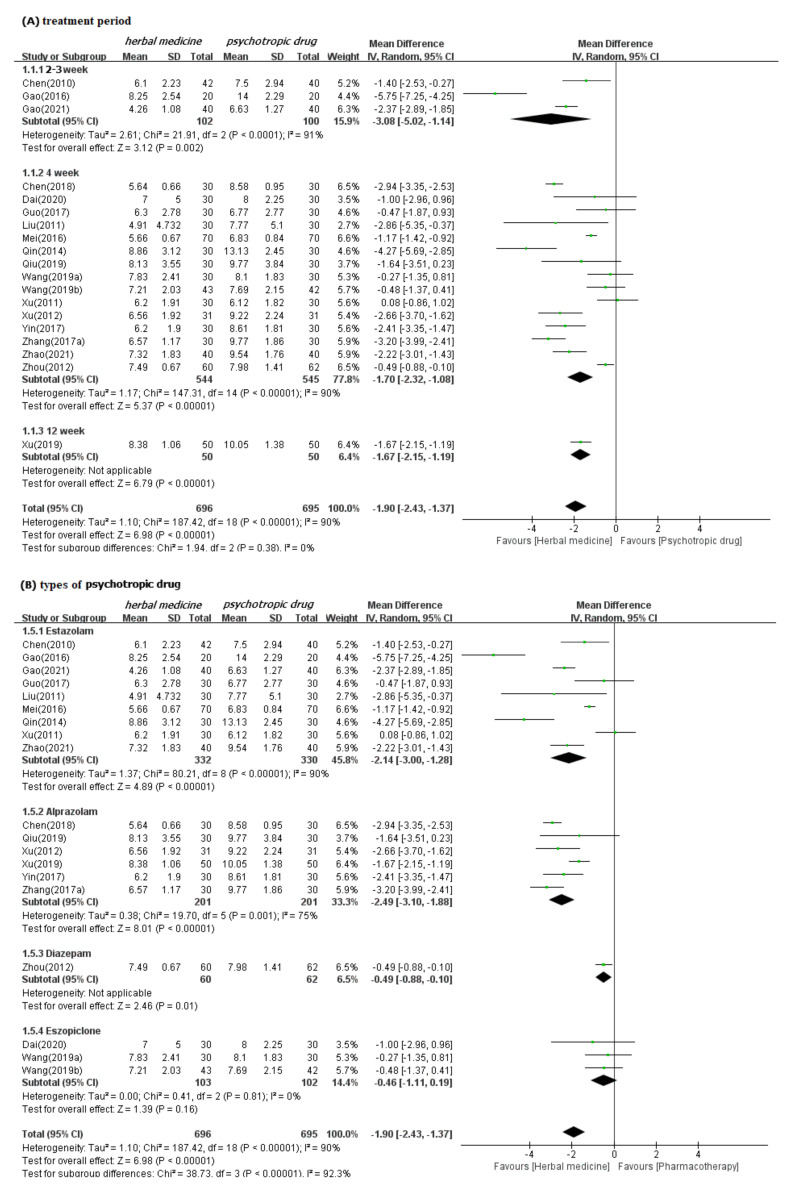
Forest plots for comparison of PSQI scores between herbal medicine and psychotropic drug groups. Subgroup analysis according to (**A**) treatment period and (**B**) types of psychotropic drug.

**Figure 4 ijerph-19-01754-f004:**
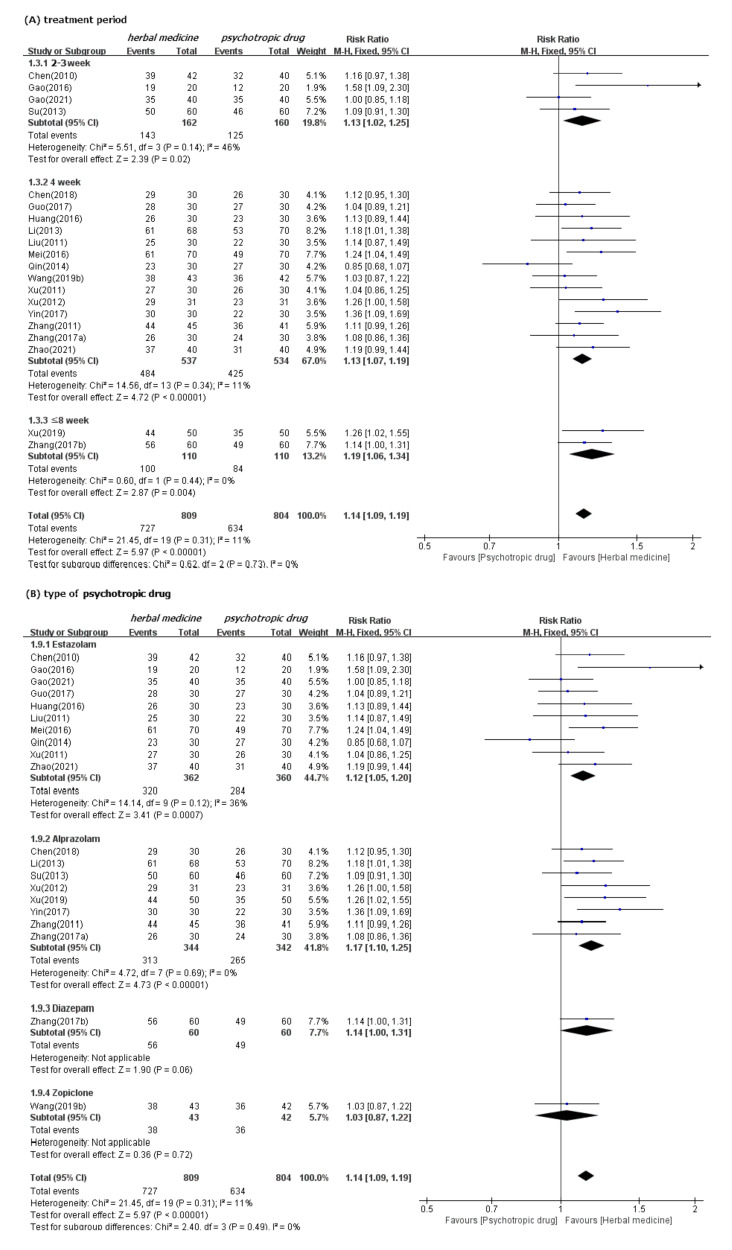
Forest plots for comparison of TER between herbal medicine and psychotropic drug groups. Subgroup analysis according to (**A**) treatment period and (**B**) types of psychotropic drug.

**Figure 5 ijerph-19-01754-f005:**
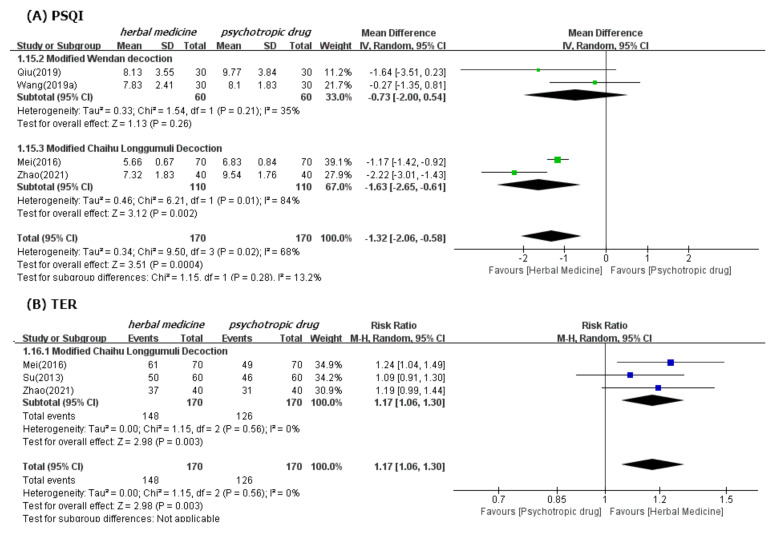
Forest plots for comparison of PSQI (**A**) and TER (**B**) between herbal medicine and psychotropic drug groups. Subgroup analysis according to Modified Wendan decoction and Chaihu Longgumuli decoction.

**Figure 6 ijerph-19-01754-f006:**
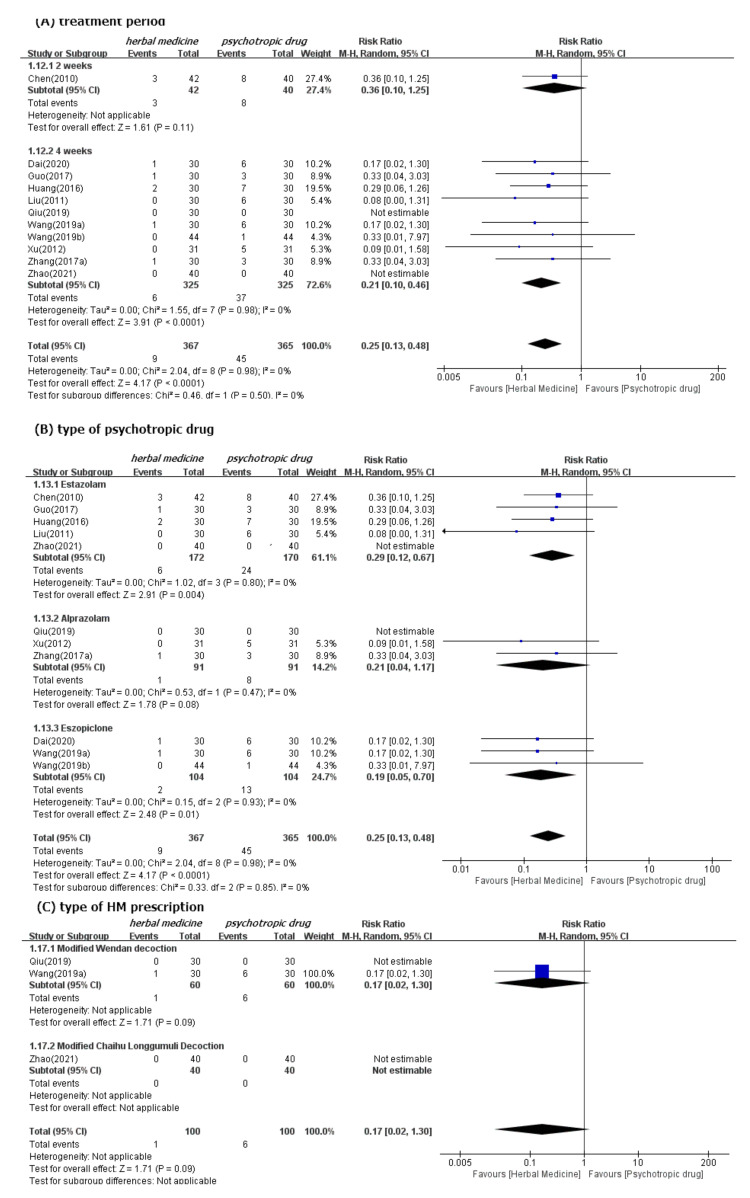
Forest plots for comparison of adverse events between herbal medicine and pharmacotherapy groups. Subgroup analysis according to (**A**) treatment period, (**B**) type of psychotropic drug, and (**C**) type of HM prescription.

**Figure 7 ijerph-19-01754-f007:**
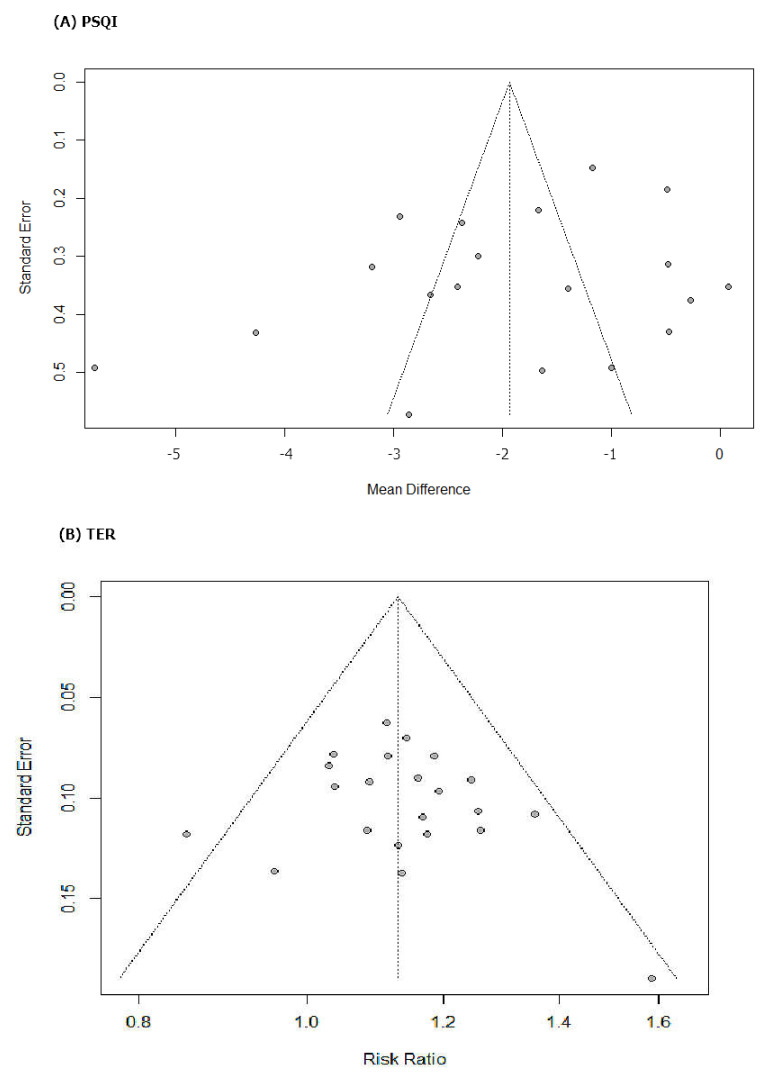
Results of the analysis of publication bias for comparison between herbal medicine and psychotropic drug. (**A**) PSQI and (**B**) TER.

**Table 1 ijerph-19-01754-t001:** The characteristics of included studies.

First Author (year)	Sample Size (Intervention:Control) (Included →Analyzed)	Mean Age (Range) (Years)	Diagnostic Tool for PSI (Severity Criteria for Inclusion)/Stroke Type (Inclusion Criteria)	Pattern Identification	(A) Treatment Intervention	(B) Control Intervention	Treatment Duration/Follow-Up	Outcome and Results (Post-Treatment)	Adverse Events
Chen (2010) [39]	82 (42:40)→82 (42:40)	(A) 58.79 ± 7.83 (B) 59.87 ± 8.32	DSM-IV/ Stroke (MRI/CT)	NA	HM	Estazolam 1 mg/day	3 weeks	① PSQI: (A) > (B) * ② TER: (A) > (B) *	(A) 3 cases (nausea & upper abdomen discomfort) (B): 8 cases (dizziness 3, sleepiness 5)
Chen (2018) [40]	60 (30:30)→60 (30:30)	(A) 72.43 ± 6.36 (B) 73.13 ± 7.11	CCMD-3 (PSQI > 7)/ Cerebral infarction or hemorrhage (MRI/CT)	Blood stasis due to qi deficiency	HM, RCS	Alprazolam 0.4 mg/day, RCS	4 weeks	① PSQI: (A) > (B) * ② TER: (A) > (B)+ ③ TCM symptom score: (A) > (B)+	NR
Dai (2020) [41]	60 (30:30:30)→60 (30:30:30)	(A) 59.87 ± 8.32 (B)−1 60.47 ± 9.21 (B)−2 59.10 ± 8.77	DSM-5 (PSQI > 7)/ Cerebral infarction (MRI/CT)	Blood deficiency and liver-heat syndrome	HM	(B)-1 Zopiclone 3.75–7.5 mg/day, (B)-2 Placebo twice/day	4 weeks	① PSQI: (A) > (B)+ ② ISI: (A) > (B)+	(A): 1 case (dispepsia) (B): 6 cases (fatigue 4, nausea 2)
Gao (2016) [42]	40 (20:20)→40 (20:20)	(A) 61.50 ± 7.25 (B) 59.90 ± 8.72	CCMD-3 (PSQI > 16)/ Cerebral infarction (MRI/CT)	Internal harassment of phlegm-heat	HM	Estazolam 2 mg/day	2 weeks	① PSQI: (A) > (B)+ ② TER: (A) > (B) *	NR
Gao (2021) [43]	80 (40:40)→80 (40:40)	NR NR	GDTICA/ Stroke (GCTNCM)	NA	HM	Estazolam 1 mg/day	2 weeks	① PSQI: (A) > (B) * ② TER: (A) > (B) * ③ ISI: (A) > (B) *	NR
Guo (2017) [44]	60 (30:30)→60 (30:30)	(A) 64.8 ± 7.6 (B) 62 ± 8.9	GDTICA/ Cerebral infarction (MRI/CT)	Liver stagnation and phlegm-heat	HM, RCS	Estazolam 1 mg/day, RCS	4 weeks/4 weeks	① PSQI: (A) > (B)+ ② TER: (A) > (B) *	(A): 1 case (diarrhea) (B): 3 cases (fatigue 2, dry mouth 1)
Huang (2016) [45]	60 (30:30)→60 (30:30)	(A) 61.4 ± 2.72 (B) 62.39 ± 3.51	CCMD-3/ Stroke (MRI/CT)	NA	HM	Estazolam 1 mg/day	4 weeks	① TER: (A) > (B) *	(A): 2 cases(stomach discomfort) (B): 7 cases (dizziness & fatigue 3, sleepiness 4)
Li (2013) [46]	138 (68:70)→138 (68:70)	(A) 69.8 ± NR (B) 67.9 ± NR	CCMD-3 (AIS > 6)/ Cerebral infarction (MRI/CT)	NA	HM, RCS	Alprazolam 0.4–0.8 mg/day, RCS	4 weeks	① TER: (A) > (B) * ② AIS: (A) > (B) *	NR
Liu (2011) [47]	60 (30:30)→60 (30:30)	(A) 66.57 ± 7.186 (B) 65.80 ± 5.845	CCMD-3/ Cerebral infarction or hemorrhage (MRI/CT)	Blood stasis	HM	Estazolam 2 mg/day	4 weeks	① PSQI: (A) > (B) * ② TER: (A) > (B)*	(A): none (B): 6 cases (dry mouse 4, headache 2)
Mei (2016) [48]	140 (70:70)→140 (70:70)	(A) 66.3 ± 6.4 (B) 65.8 ± 7.7	GDTICA/ Cerebral infarction (MRI/CT)	NA	HM	Estazolam 1 mg/day	4 weeks	① PSQI: (A) > (B)+ ② TER: (A) > (B)+	NR
Qin (2014) [49]	60 (30:30)→60 (30:30)	(A) 62.3 ± 10.53 (B) 63.85 ± 9.78	CCMD-3, ICD-10/ Cerebral infarction or hemorrhage (MRI/CT)	NA	HM, RCS	Estazolam 1–2 mg/day, RCS	4 weeks	① PSQI: (A) > (B) * ② TER: (A) > (B)*	NR
Qiu (2019) [50]	60 (30:30)→60 (30:30)	(A) 62.57 ± 6.40 (B) 61.07 ± 7.52	GDTICA/ Cerebral infarction (MRI/CT)	Internal harassment of phlegm-heat	HM, RCS	Alprazolam 0.4 mg/day, RCS	4 weeks	① PSQI: (A) > (B) *	none (B): none
Su (2013) [51]	120 (60:60)→120 (60:60)	(A) NR (B) NR	CCMD-3/ Cerebral infarction (MRI/CT)	NA	HM, RCS	Alprazolam 1 mg/day, RCS	2 weeks	① TER: (A) > (B) *	NR
Wang (2019a) [52]	60 (30:30)→NR	(A) 60.67 ± 8.63 (B) 61 ± 8.67	CCMD-3/ Cerebral infarction (MRI/CT)	Internal harassment of phlegm-heat	HM, RCS	Zopiclone 7.5 mg/day, RCS	4 weeks	① PSQI: N.S ② PSG: N.S (all parameters)③ TCM symptom score: (A) > (B)+	(A): none(B): 1 case (sleepiness)
Wang (2019b) [53]	88 (44:44)→85 (43:42)	(A) 61.89 ± 8.56 (B) 62.02 ± 6.31	CCMD-3/ Cerebral infarction or hemorrhage (MRI/CT)	Heart-kidney non-interaction	HM, RCS	Zopiclone 3 mg/day, RCS	4 weeks	① PSQI: N.S② TER: N.S③ TCM symptom score: (A) > (B) *	(A): 1 case (diarrhea) (B): 6 cases (dry mouse 4, headache 2)
Xu (2011) [54]	60 (30:30)→60 (30:30) 30	(A) 66.1 ± 7.8 (B) 65.8 ± 7.2	DSM(PSQI > 7)/ Cerebral infarction (MRI/CT)	Kidney yin deficiency and blood stasis	HM, RCS	Estazolam 1 mg/day, RCS	4 weeks	① PSQI: N.S② TER: N.S ③ TCM symptom score: (A) > (B) *	TESS: (A) > (B)*
Xu (2012) [55]	62 (31:31)→62 (31:31)	(A) 72.2 ± 4.8 (B) 70.2 ± 3.9	CCMD-3 (PSQI > 7)/ Cerebral infarction(MRI/CT)	NA	HM, RCS	Alprazolam 0.8 mg/day, RCS	4 weeks	① PSQI: (A) > (B)+ ② TER: (A) > (B) *	(A): none (B): 5 cases (dizziness)
Xu (2019) [56]	100 (50:50)→100 (50:50)	(A) 66.32 ±4.37 (B) 66.67 ± 4.42	CCMD-3(SSQ > 12)/ Cerebral infarction (MRI/CT)	Liver stagnation and blood stasis	HM, RCS	Alprazolam 1mg/day, RCS	12 weeks	① PSQI: (A) > (B) * ② TER: (A) > (B) * ③ SSQ: (A) > (B) *	NR
Yin (2017) [57]	60 (30:30)→60 (30:30)	(A) 64.2 ± 6.3 (B) 62.8 ± 6.9	GPCRNDTCM/ Cerebral infarction or hemorrhage (MRI/CT)	Kidney yin deficiency and blood stasis	HM, RCS	Alprazolam 2 mg/day, RCS	4 weeks	① PSQI: (A) > (B)+ ② TER: (A) > (B) *	NR
Zhang (2011) [58]	86 (45:41)→86 (45:41)	(A) NR (B) NR	CCMD-3/ Cerebral infarction or hemorrhage (MRI/CT)	NA	HM	Alprazolam 0.4 mg/day	4 weeks	① TER: (A) > (B) *	NR
Zhang (2017a) [60]	64 (32:32)→60 (30:30)	(A) 53.50 ±9.52 (B) 54.10 ± 9.78	GDTICA (PSQI > 7)/ Cerebral infarction or hemorrhage (MRI/CT)	Disturbing heart due to liver burning	HM, RCS	Alprazolam 0.5–1.5 mg/day, RCS	4 weeks	① PSQI: (A) > (B) * ② TER: N.S③ TCM symptom score: (A) > (B) *	(A): 1 case (stomach discomfort) (B): 3 cases (dizziness 2, fatigue 1)
Zhang (2017b) [59]	120 (60:60)→120 (60:60)	(A) NR (B) NR	ICSD-2/ Cerebral infarction (MRI/CT)	NA	HM	Diazepam 5~10 mg/day	8 weeks	① TER: (A) > (B) *	NR
Zhao (2021) [61]	80 (40:40)→80 (40:40)	(A) 52.7 ± 6.1 (B) 53.5 ± 5.9	DSM-5/ Cerebral infarction or hemorrhage (MRI/CT)	Liver stagnation and blood deficiency	HM	Estazolam 1 mg/day	4 weeks/4 weeks	① PSQI: (A) > (B) * PSQI(f/u): (A) > (B) * ② TER: (A) > (B) * ③ SS-QOL: A) > (B) * SS-QOL (f/u): (A) > (B) *	none (B): none
Zhou (2012) [62]	142 (60:62)→NR	(A) 60.19 ± 4.80 (B) 59.72 ± 10.71	CCMD-3/ Cerebral infarction or hemorrhage (MRI/CT)	NA	HM	Diazepam 2 mg/day	4 weeks/4 weeks	① PSQI: N.S PSQI (f/u): (A) > (B) *	NR

Note: ‘*’ and ‘+’ mean significant differences between two groups, *p *< 0.05 and *p* < 0.01, respectively. ‘N.S’ means no significant difference between two groups, *p* > 0.05. AIS = Athens Insomnia Scale; CCMD = Chinese Classification of Mental Disorders; CT = Computed Tomography; DSM = the Diagnostic and Statistical Manual of Mental Disorders, International Classification of Diseases; GDTICA = Guideline for the Diagnosis and Treatment of Insomnia in Chinese Adults; GPCRNDTCM = the Guiding Principles for Clinical Research on New Drugs of Traditional Chinese Medicine; HM = Herbal Medicine; ICSD = International Classification of Sleep Disorders; ISI = Insomnia Severity Index; MRI = Magnetic Resonance Imaging; NA = Not Applicable; NR = Not Reported; PSQI = Pittsburgh Sleep Quality Index; RCS = Routine Care for Stroke; SS-QOL = Stroke Specific Quality of Life Scale; TER = Total Effective Rate; TESS = Treatment-Emergent Signs and Symptoms.

**Table 2 ijerph-19-01754-t002:** The qualities of evidence regarding each outcome and subgroup.

Outcomes		No. Participants (RCTs)	Anticipated Absolute Effects (95% CI)		Quality of Evidence (GRADE)
			Risk with Pharmacotherapy	Risk with Herbal Medicine	
PSQI	Total	1391 (19)	-	MD 1.9 lower (2.43 to 1.37 lower)	⊕⊕⊕◯ MODERATE ^a^
Subgroup 1	2–3 week	202 (3)	-	MD 3.08 lower (5.02 to 1.14lower)	⊕⊕◯◯ LOW ^ab^
	4 week	1089 (15)	-	MD 1.7 lower (2.32 to 1.08 lower)	⊕⊕⊕◯ MODERATE ^a^
	12 week	100 (1)	-	MD 1.67 lower (2.15 to 1.19 lower)	⊕⊕◯◯ LOW ^ab^
Subgroup 2	Estazolam	662 (9)	-	MD 2.14 lower (3 to 1.28 lower)	⊕⊕⊕◯ MODERATE ^a^
	Alprazolam	402 (6)	-	MD 2.49 lower (3.1 to 1.88 lower)	⊕⊕⊕◯ MODERATE ^a^
	Diazepam	122 (1)	-	MD 0.49 lower (0.88 to 0.1 lower)	⊕⊕◯◯ LOW ^ab^
	Eszopiclone	205 (3)	-	MD 0.46 lower (1.11 lower to 0.19 higher)	⊕⊕◯◯ LOW ^ac^
Subgroup 3	Wendan decoction	120 (2)		MD 0.73 lower (2.00 lower to 0.54 higher)	⊕⊕◯◯ LOW ^ac^
	Chaihu Longgumuli Decoction	220 (2)		MD 1.63 lower (2.65 to 0.61 lower)	⊕⊕◯◯ LOW ^ab^
PSQI (4 weeks f/u)	Total	262 (3)		MD 3.08 lower (3.52 to 2.64 lower)	⊕⊕◯◯ LOW ^ab^
TER	Total	1673 (21)	783 per 1.000	110 more per 1.000 (70 to 149)	⊕⊕⊕◯ MODERATE ^a^
Subgroup 1	2–3 week	322 (4)	781 per 1.000	102 more per 1.000 (16 to 195)	⊕⊕◯◯ LOW ^ac^
	4 week	1071 (14)	796 per 1.000	103 more per 1.000 (56 to 151)	⊕⊕⊕◯ MODERATE ^a^
	≤8 week	220 (2)	764 per 1.000	145 more per 1.000 (46 to 260)	⊕⊕◯◯ LOW ^ab^
Subgroup 2	Estazolam	722 (10)	775 per 1.000	108 more per 1.000 (54 to 170)	⊕⊕⊕◯ MODERATE ^a^
	Alprazolam	686 (8)	775 per 1.000	132 more per 1.000 (77 to194)	⊕⊕⊕◯ MODERATE ^a^
	Diazepam	120 (1)	817 per 1.000	114 more per 1.000 (0 to 253)	⊕⊕◯◯ LOW ^ac^
	Eszopiclone	145 (2)	833 per 1.000	0 fewer per 1.000 (108 fewer to 133 more)	⊕⊕◯◯ LOW ^ac^
Subgroup 3	Chaihu Longgumuli Decoction	340 (3)	741 per 1.000	126 more per 1.000 (44 to 222)	⊕⊕◯◯ LOW ^ab^
Adverse effects	Total	792 (12)	114 per 1.000	89 fewer per 1.000 (100 to 65)	⊕⊕⊕◯ MODERATE ^a^
Subgroup 1	2 week	82 (1)	200 per 1.000	128 fewer per 1.000 (180 fewer to 50 more)	⊕⊕◯◯ LOW ^ac^
	4 week	710 (11)	104 per 1.000	84 fewer per 1.000 (95 to 60)	⊕⊕⊕◯ MODERATE ^a^
Subgroup 2	Estazolam	402 (6)	120 per 1.000	89 fewer per 1.000 (107 to 48)	⊕⊕⊕◯ MODERATE ^a^
	Alprazolam	182 (3)	88 per 1.000	72 fewer per 1.000 (85 to 3)	⊕⊕◯◯ LOW ^ac^
	Eszopiclone	208 (3)	125 per 1.000	101 fewer per 1.000 (119 to 39)	⊕⊕◯◯ LOW ^ab^
Subgroup 3	Wendan decoction	120 (2)	100 per 1.000	83 fewer per 1.000 (98 fewer to 30 more)	⊕⊕◯◯ LOW ^ac^
	Chaihu Longgumuli Decoction	80 (1)	0 per 1.000	NA	⊕⊕◯◯ LOW ^ac^

Note: CI = Confidence of Interval; MD = Mean Deference; NA = Not Applicable PSQI = Pittsburgh Sleep Quality Index; TER = Total Effective Rate. ^a^: The included study(ies) had a risk of selection and performance bias. ^b^: Sample size < 400 ^c^: Sample size < 400, the 95% confidence interval overlapped with no effect.

**Table 3 ijerph-19-01754-t003:** Results of Egger’s regression of the analysis of publication bias for comparison of total effective rate between herbal medicine and pharmacotherapy groups (**a**) PSQI and (**b**) TER.

(a) PSQI.
Test Result:
t	df	*p*-Value	□
−1.45	17	0.1656	□
Sample Estimates:
bias	se.bias	intercept	se.intercept
−3.6094	2.4915	−0.6967	0.7193
(b) TER

Test Result:
t	df	*p*-Value	□
0.83	20	0.4157	□
Sample Estimates:
bias	se.bias	intercept	se.intercept
0.7602	0.9147	0.0516	0.0872

Note: PSQI, Pittsburgh Sleep Quality Index, TER = Total Effective Rate.

## Data Availability

The original contributions presented in the study are included in the article/Appendix A, further inquiries can be directed to the corresponding author.

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
