# Peer review of "Traditional East Asian Herbal Medicine for Post-Stroke Insomnia: A Systematic Review and Meta-Analysis of Randomized Controlled Trials"

_ijerph, 2022, doi:10.3390/ijerph19031754_

Round 1

Reviewer 1 Report

This is a very unique study. The concept was previously evaluated but failed to generate strong analysis and/or evidence. 

The authors did an excellent job with systematic review and meta-analysis. Very well written article. Even after through review, I do not have any significant comments.  I wish table 1 can have more clarity, it looks somewhat crowded. Thank you for drawing a clear conclusion. 

Great study. Thank you for your hard work. 

Author Response

Comment 1:

This is a very unique study. The concept was previously evaluated but failed to generate strong analysis and/or evidence. The authors did an excellent job with systematic review and meta-analysis. Very well written article. Even after through review, I do not have any significant comments.

 I wish table 1 can have more clarity, it looks somewhat crowded. Thank you for drawing a clear conclusion. Great study. Thank you for your hard work.

Response 1:           

Thank you for your warm comment. We have edited table 1 for clarity as your comment and added some note.

Reviewer 2 Report

This is a systematic review which investigates the effects of herbal medicine for post-stroke insomnia. Despite the systematic search and assessment process, considerable caveats with regard to risk of bias assessment need to be revised and conclusions should not be exaggerated. Please check minor typos such as ‘date synthesis’ (data synthesis) or ‘zopidem’ (zolpidem). 

Please check the below comments: 

#1. Abstract conclusions: this is not based on the review results. Due to high or unclear risk of bias, apparent beneficial effects of HM should not be accepted as observed. Please integrate the review findings with regard to the bias assessment when drawing conclusions. 

#2. ROB assessment 2.5; please specify rationale of selecting mean age and baseline insomnia level for factors associated with other risk of bias.

#3. Fixed or Random effects model 2.6.3; Please be aware the post-hoc selection of analysis model (fixed or random) is strongly discouraged. Selection or fixed-effect or random-effects model depends on assumption of true-effect distribution (a single-true effect or multiple-true effects) and should be pre-specified, rather than chosen by heterogeneity as post-hoc manner. Authors may refer to Borenstein et al. (Introduction to Meta-Analysis, 1st ed. 2009; Chap 13. Fixed-effect versus random-effects models) for further guidance.

#4. Sensitivity analysis 2.6.5; please specify the categorization of duration of treatment  and type of psychotropic drugs for the sensitivity analysis. 

#5. Subgroup analysis 2.6.4; please specify which domains of risk of bias will serve as criteria for subgroup analysis. (Selection bias, performance bias or overall bias?) I am not sure what is meant by studies ‘numerically distant from the rest data’. 

#6.  GRADEing 2.6.7; please specify the criteria of upgrading and downgrading of evidence as well as provide brief summary of GRADE process. 

#7. ROB in studies 3.3; Selective outcome reporting assessment; There is empirical evidence that selective outcome reporting bias can occur even the outcomes defined in the methods were all reported in the results unless the review is prospectively registered. [ref] Rating this domain as ‘unclear’ may be appropriate. 

#8. Methods; method of safety assessment was not reported. Please specify how authors analyzed safety of herbal medicine over comparator interventions. 

#9. Safety assessment results 3.4.2; Please consider descriptive summarization of adverse events and safety issues as well as meta-analysis. 

#10. GRADE assessment (3.5) and Table 2; please add footnotes for Table 2 which illustrate reasons of upgrading / downgrading the certainty of evidence. (Also in the manuscript)

#11. 4.1 Summary of evidence: I suppose the major finding of this review (based on the review findings) will be that ‘there is moderate level of evidence that herbal medicine intake is associated with the mean reduction of 1.9 points of PSQI compared with control medication (MD -1.9 points, 95% CI -1.37 to -2.43).’ Current descriptions can be improved by focusing on the findings of primary outcomes of this review and the overall interpretation of what authors have found. Please also consider re-writing the discussion which concentrates the primary findings of this review and its interpretation in context of previous evidences. 

#12. Clinical implications (4.2); Please specify the accurate nomenclature of Z-drug. “HM may be considered a better alternatives to benzodiazepines than zolpiclone for PSI’ Findings of subgroup analyses should be regarded as exploratory, as the nature of subgroup analysis is observational. (there prone to selection bias) This interpretation seems too conclusive, in my view, given the high risk of bias and observational subgroup analysis. 

#13. Strengths and limitations (4.3): Given the high or unclear risk of selection and performance bias, the compelling evidence of HM for PSI is unlikely, in my view. Please integrate the bias assessment into the review findings for valid interpretation of results. Publication bias is not associated with generalizability of review findings, so necessity of further studies in countries other than China should be mentioned based on different rationale. I am not sure multi-target mechanisms can be mentioned in the limitation paragraph as this issue is not addressed during the review. 

#14. Conclusion: the conclusions should be based on overall interpretation of review findings and bias assessment results (GRADE). ‘HM appears to be a promising CAM therapy’ —> This is ambiguous description, and needs to be removed or revised. Please summarize the review findings based no the certainty of evidence, rather than providing vague impression on the results. 

Author Response

This is a systematic review which investigates the effects of herbal medicine for post-stroke insomnia. Despite the systematic search and assessment process, considerable caveats with regard to risk of bias assessment need to be revised and conclusions should not be exaggerated. Please check minor typos such as ‘date synthesis’ (data synthesis) or ‘zopidem’ (zolpidem).

Response:

Thank you for your insightful feedbacks. We did our best to address the weaknesses of this paper in accordance with your comments and suggestions. And we changed typos such as ‘date synthesis’ (data synthesis) or ‘zopidem’ (zolpidem).

Comment 1:

Abstract conclusions: this is not based on the review results. Due to high or unclear risk of bias, apparent beneficial effects of HM should not be accepted as observed. Please integrate the review findings with regard to the bias assessment when drawing conclusions.

Response 1:           

We have changed conclusions in the abstract based on the bias assessment as your comment (marked in yellow/ Abstract).

“The findings of this review indicate that the use of HM as a monotherapy may have potential benefits in PSI treatment when administered as an alternative to conventional medications. However, considering the methodological quality of the included RCTs, we were uncertain of the clinical evidence. Further well-designed RCTs are required to confirm these findings.”

Comment 2:

ROB assessment 2.5; please specify rationale of selecting mean age and baseline insomnia level for factors associated with other risk of bias.

Response 2:           

Thank you for your comment. We added the rationale to assess baseline insomnia level between experimental and control group (marked in yellow). The rationale of selecting mean age is not specific value but baseline imbalances between experimental and control group to assess other potential bias categories.

“Other potential bias categories were assessed by particularly the emphasis on baseline imbalances between experimental and control group such as participant characteristics including the mean age and baseline insomnia level based on PSQI, Insomnia Severity Index (ISI) and the Athens Insomnia Scale (AIS)”

Comment 3:

Fixed or Random effects model 2.6.3; Please be aware the post-hoc selection of analysis model (fixed or random) is strongly discouraged. Selection or fixed-effect or random-effects model depends on assumption of true-effect distribution (a single-true effect or multiple-true effects) and should be pre-specified, rather than chosen by heterogeneity as post-hoc manner. Authors may refer to Borenstein et al. (Introduction to Meta-Analysis, 1st ed. 2009; Chap 13. Fixed-effect versus random-effects models) for further guidance.

Response 3:           

As the reviewer pointed out, selection or fixed-effect or random-effects model depends on assumption of true-effect distribution (a single-true effect or multiple-true effects) and should be pre-specified, rather than chosen by heterogeneity as post-hoc manner. We have pre-specified assumption of true-effect distribution to select fixed-effect or random-effects model (marked in yellow). And according to this change, we have changed Figure 4,5 using random-effects model and specific values in the results (marked in yellow).

“We assumed that the true effect size varies from one study to the next, and that the studies in our analysis represent a random sample of effect sizes that could have been observed, because there are significant clinical heterogeneity across included studies. Therefore the data were pooled using a random-effects model regardless of the heterogeneity. However, the data were pooled using fixed-effects model when the number of studies included in the meta-analysis was extremely small, meaning that the estimate of the between-study variance lacked precision.”

Comment 4:

Subgroup analysis 2.6.4; please specify the categorization of duration of treatment and type of psychotropic drugs for the sensitivity analysis.

Response 4:           

We have added as below (marked in yellow).

“We conducted a subgroup analysis considering the duration of treatment, the type of psychotropic drug and HM prescription. The duration of treatment was classified as 2–3 weeks, 4 weeks, and > 8 weeks, respectively, and the psychotropic drugs alprazolam, estazolam, and zopiclone were investigated.”

Comment 5:

Sensitivity analysis 2.6.5; please specify which domains of risk of bias will serve as criteria for subgroup analysis. (Selection bias, performance bias or overall bias?) I am not sure what is meant by studies ‘numerically distant from the rest data’.

Response 5:           

We have added as below (marked in yellow).

“Additionally, we conducted sensitivity analysis by excluding the (1) studies that were rated as high RoB based on selection bias and (2) studies that were numerically distant from the rest data (MD and RR of each study result is -4 points and ≥ 1.5, respectively).”

Comment 6:

GRADEing 2.6.7; please specify the criteria of upgrading and downgrading of evidence as well as provide brief summary of GRADE process.

Response 6:           

We have added as below (marked in yellow).

“The GRADE process begins with asking an explicit question, including specification of all important outcomes. After the evidence is collected and summarized, GRADE provides explicit criteria for rating the quality of evidence that include study design, risk of bias, imprecision, inconsistency, indirectness, and magnitude of effect. The GRADE system rates the quality of evidence for each outcome, from a rating of HIGH to VERY LOW. GRADE starts with a baseline rating of HIGH for RCTs, and LOW for non-RCTs. This baseline rating can then be adjusted (downgraded or, less commonly, upgraded) after considering 8 assessment criteria. Reasons to downgrade the evidence quality are 1) Risk of Bias, 2) Inconsistency, 3) Indirectness, 4) Imprecision and 5) Publication Bias. And reasons to upgrade the evidence quality are 6) Large Magnitude of Effect, 7) Dose Response, 8) Effect of all plausible confounding factors would be to reduce the effect (where an effect is observed) or suggest a spurious effect (when no effect is observed). After then, we finally did make a judgement about quality based on these.”

Comment 7:

ROB in studies 3.3; Selective outcome reporting assessment; There is empirical evidence that selective outcome reporting bias can occur even the outcomes defined in the methods were all reported in the results unless the review is prospectively registered. [ref] Rating this domain as ‘unclear’ may be appropriate.

Response 7:           

Thank you for your comment. We have rated the domain of selective outcome reporting assessment as ‘unclear’ (Figure 2) except one study [41] which is prospectively registered. And we changed sentences in 3.3. RoB in studies (marked in yellow).

“We evaluated all studies except one [41] as having unclear ROB in the selective reporting domain because the protocol of each study was not prospectively registered.”

Comment 8:

Methods; method of safety assessment was not reported. Please specify how authors analyzed safety of herbal medicine over comparator interventions.

Response 8:           

We have added as below (marked in yellow).

“The safety assessment evaluated the incidence of AEs between the experimental intervention and control intervention groups”

Comment 9:

Safety assessment results 3.4.2; Please consider descriptive summarization of adverse events and safety issues as well as meta-analysis.

Response 9:           

We have added as below (marked in yellow).

“The AEs associated with HM included upper abdominal or stomach discomfort (5 cases), diarrhea (2 cases), nausea (1 case), and dispepsia (1 case). The AEs associated with psychotropic drugs included dizziness (13 cases), fatigue (10 cases), sleepiness (10 cases), dry mouth (9 cases), headache (4 cases), and nausea (2 cases). All AEs associated with HM included gastrointestinal symptoms, and the AEs associated with psychotropic drugs included various symptoms, such as neurolocial and general symptoms.”

Comment10:

GRADE assessment (3.5) and Table 2; please add footnotes for Table 2 which illustrate reasons of upgrading / downgrading the certainty of evidence. (Also in the manuscript)

Response 10:         

Thank you for your comment. We have put footnotes and deleted Comments Imprecision in Table 2(marked in yellow).

Note. CI = Confidence of Interval; MD = Mean Deference; NA = Not Applicable PSQI = Pittsburgh Sleep Quality Index; TER = Total Effective Rate

a: The included study(ies) had a risk of selection and performance bias.

b: Sample size < 400

c: Sample size < 400, the 95% confidence interval overlapped with no effect.

“The main reason for this was the high ROB in the RCTs considered in each meta-analysis. The included study(ies) had a risk of selection and performance bias. Moreover, most findings in subgroup analysis were judged to have low precision because they had wide CIs and did not satisfy the optimal sample size.”

Comment 11:

4.1 Summary of evidence: I suppose the major finding of this review (based on the review findings) will be that ‘there is moderate level of evidence that herbal medicine intake is associated with the mean reduction of 1.9 points of PSQI compared with control medication (MD -1.9 points, 95% CI -1.37 to -2.43).’ Current descriptions can be improved by focusing on the findings of primary outcomes of this review and the overall interpretation of what authors have found. Please also consider re-writing the discussion which concentrates the primary findings of this review and its interpretation in context of previous evidences.

Response 11:         

Thank you for your kindly suggestion and comment. Based on your comment, we re-writing the discussion which concentrates the primary findings of this review and its interpretation in context of previous evidences (4.2. Interpretation in context of previous evidences).

“4.2. Interpretation in context of previous evidences

“Our meta-analysis results provide limited evidence of HM as monotherapy being beneficial in PSI treatment compared to psychotropic drugs, especially within the first 2–4 weeks of treatments. This finding is consistent with the findings of previous systematic reviews comparing overall HM with benzodiazepine drugs for primary insomnia or insomnia disorder [64]. In the previous reviews, HM was found to be more effective than benzodiazepine drugs in reducing PSQI scores (MD: −1.94, 95% CI: −2.45 to −1.43). In our meta-analysis, the mean reduction of 1.9 points in PSQI score was below 3 points, indicating a minimal clinically important difference in patients with primary insomnia [65]. Although the TER in the HM group, which may reflect changes in insomnia in the clinical setting, was higher than the TER in the medication group, we were uncertain of this evidence, because its quality was not high.

The effect sizes based on the PSQI of each HM prescription for treating insomnia in previous systematic reviews have differed. The effect size of the Chaihu-Longgu-Muli decoction (MD = -2.80, 95% CI -5.48, -0.13, P = .04) [66] was larger than that of our results, while the results of Xiao-yao-san (MD: -1.82; 95% CI -2.39 to -1.24; P < .001) [67] was similar to ours. However, those of Shumian capsules (MD = -0.50, 95% confidence interval [CI] = [-0.78, -0.22], P = .0005) [68] and the Banxia Formulae (MD = -1.05, 95% CI -1.63 to -0.47) [69] were smaller than our results. The population of these studies did not include patients with insomnia accompanied by diseases such as PSI, but rather, patients with primary insomnia or insomnia disorder were evaluated. Thus, to interpret our primary results, we must consider the heterogeneity and different effect sizes of each HM prescription.”

Comment12:

Clinical implications (4.2); Please specify the accurate nomenclature of Z-drug. “HM may be considered a better alternatives to benzodiazepines than zolpiclone for PSI’ Findings of subgroup analyses should be regarded as exploratory, as the nature of subgroup analysis is observational. (there prone to selection bias) This interpretation seems too conclusive, in my view, given the high risk of bias and observational subgroup analysis.

Response 12:         

Thank you for your comment. We specify the accurate nomenclature of Z-drug. (marked in yellow / 1. Introduction). And we deleted too conclusive this sentence (“This finding indicates that HM may be considered a better alternative to benzodiazepines than zolpiclone for PSI.”) according to your suggestion.

From our results, HM shows better effect than zolpiclone and similar effect when compared with benzodiazepines (especially estazolam and alprazolam). This finding indicates that HM may be considered a better alternative to benzodiazepines than zolpiclone for PSI.

“Zolpidem, zopiclone, and zaleplon are non-benzodiazepine drugs used in the treatment of insomnia and commonly referred to as the “Z-drugs because the generic names of two of the three currently approved agents in the U.S. begin with the letter Z.”

Comment13:

Strengths and limitations (4.3): Given the high or unclear risk of selection and performance bias, the compelling evidence of HM for PSI is unlikely, in my view. Please integrate the bias assessment into the review findings for valid interpretation of results. Publication bias is not associated with generalizability of review findings, so necessity of further studies in countries other than China should be mentioned based on different rationale. I am not sure multi-target mechanisms can be mentioned in the limitation paragraph as this issue is not addressed during the review.

Response 13:         

Thank you for your comment. We changed sentence as below.

“our findings might help provide limited evidence for optimal recommendations of HM as an alternative to benzodiazepines or Z-drugs in clinical practice for the treatment of PSI.”

We deleted this sentence based on in your suggestion. And we mentioned the necessity of further studies based on different rationale.

“Although relatively many studies have been examined, we find no evidence of publication bias.”

“In the future, we need to conduct further studies in countries other than China with different rationales.”

We deleted this sentence regading this issue is not addressed during the review.

Seventh, HM exerted effects through multi-component, multi-target, and multi-pathway mechanisms [29] and is widely used in treating stroke in EATM [74,75]. In a recent review (Chai-Hu-Jia-Long-Gu-Mu-Li-Tang), HM monotherapy significantly improved both depression and neurological functions [76].

Comment14:

Conclusion: the conclusions should be based on overall interpretation of review findings and bias assessment results (GRADE). ‘HM appears to be a promising CAM therapy’ —> This is ambiguous description, and needs to be removed or revised. Please summarize the review findings based no the certainty of evidence, rather than providing vague impression on the results.

Response 14:         

Thank you for your comment. We deleted ambiguous sentence and changed sentence based on in your suggestion.

‘HM appears to be a promising CAM therapy’

“The findings of this review indicate that the use of HM as a monotherapy may have potential benefits in PSI treatment when administered as an alternative to conventional medications such as benzodiazepines and eszopiclone. However, considering the methodological quality of the included RCTs and the fact that no placebo-controlled trials were conducted, we were uncertain of the clinical evidence. Therefore, further well-designed RCTs must confirm the above findings.”

Reviewer 3 Report

The present manuscript is a systematic review and meta-analysis evaluating the efficacy and safety of traditional East Asian herbal medicine for post stroke insomnia. The research methodology and analysis results are well written in the text. However, some revisions are required to be considered for publication.

<Title>

In this study, it seems that analysis was conducted on herbal medicine based on traditional East Asian medicine among various herbal medicines. Therefore, I think it would be more appropriate to title it traditional East Asian herbal medicine instead of herbal medicine.

<Methods & Results>

In this study, a subgroup analysis was performed according to treatment period and control mediciation in meta-analysis, but subgroup analysis according to the type of herbal medicine used was not performed. Therefore, it would be better if subgroup analysis by types of herbal medicine prescription was conducted. It appears that the authors did not conduct an analysis because the same regimen had never been used. However, if you look at supplementary material 4, you can see that various Wendan decoction variants are frequently used. Wendan decoction is a prescription that has been traditionally mainly used for insomnia, and it would be good if a subgroup analysis on it proceeds.

<Discussion>

The authors explained the mechanism of action of herbal medicines. However, it did not describe which herbs correspond to each mechanism. I think it will be more meaningful if authors add specific mechanisms for each prescription used in the RCT that were analyzed in this study or for herbs that have been frequently used.

Author Response

Comment 1:

<Title> In this study, it seems that analysis was conducted on herbal medicine based on traditional East Asian medicine among various herbal medicines. Therefore, I think it would be more appropriate to title it traditional East Asian herbal medicine instead of herbal medicine.

Response 1:           

Thank you for your suggestion. We have revised the title as ‘traditional East Asian herbal medicine’.

Comment 2:

<Methods & Results> In this study, a subgroup analysis was performed according to treatment period and control mediciation in meta-analysis, but subgroup analysis according to the type of herbal medicine used was not performed. Therefore, it would be better if subgroup analysis by types of herbal medicine prescription was conducted. It appears that the authors did not conduct an analysis because the same regimen had never been used. However, if you look at supplementary material 4, you can see that various Wendan decoction variants are frequently used. Wendan decoction is a prescription that has been traditionally mainly used for insomnia, and it would be good if a subgroup analysis on it proceeds.

Response 2:           

Thank you for your careful comment. We have added new subgroup analysis according to the types of herbal medicine prescription such as Wendan decoction [50,52] and Chaihu Longgumuli Decoction [61] (Figure 5). We also have revised subgroup analysis 2.6.4 in the Method, Figure 5 and GRADE of table 2(marked in yellow).

Comment 3:

<Discussion>The authors explained the mechanism of action of herbal medicines. However, it did not describe which herbs correspond to each mechanism. I think it will be more meaningful if authors add specific mechanisms for each prescription used in the RCT that were analyzed in this study or for herbs that have been frequently used.

Response 3:           

I did describe the name of herbs correspond to each mechanism. And we added table of frequencies of usage in each HM prescriptions (Supplementary 5) and the mechanisms of major each herbs (marked in yeollow/ manuscript).

4.4. the underlying mechanism of HM

“Numerous preclinical studies have examined the underlying mechanism of various HM’s action: (1) Suanzaoren decoction could induce sleep through anxiolytic-like activities and regulation of neurotransmitter levels, for example, glutamate and γ-aminobutyric acid (GABA) [74]. (2) Suanzaoren decoction regulated the circadian rhythm by regulating the mRNA and protein expressions in the suprachiasmatic nucleus [75]. (3) Chaihu-Longgu-muli decoction inhibits adrenocorticotropic hormone and corticosterone, both of which may be involved in regulating insomnia [76].  (4) Jiaotai Pill alleviated insomnia by regulating the monoaminergic system and activities of organic cation transporters in the hypothalamus and peripheral organs [77]. (5) Jiaotai Pill might promote the repair of intestinal epithelial barriers and suppress the systemic inflammation and cognitive impairment in sleep-deprived rats [78].

Sanjoinine A, JuA, and flavonoids in the water extract of zizyphi spinosae semen, the most commonly used herb in our studies, exerted sedative-hypnotic effects by increasing chloride influx and over-expression of α- and γ-subunit GABA receptors in the GABAergic and serotonergic systems [79]. Furthermore, the combination of zizyphi spinosae semen and rhizoma salviae miltiorrhizae showed a significant synergistic effect (p<0.05) in decreasing sleep latency and increasing sleeping time in a mouse model using the pentobarbital-induced sleep method [80]. The sedative-hypnotic effect of HM (herbal pair of semen ziziphi spinosae and radix polygalae) was possibly related to the adjustment of the neurotransmitters 5-hydroxytryptamine, norepinephrine, and dopamine in the total protein of mouse brain tissue [81]. Poria cocos (hoelen) enhances pentobarbital-induced sleeping behaviors via GABAergic mechanisms in rodents [82]. The central inhibitory effects of acori graminei rhizoma were probably affected by an action on the central dopamine and GABA(A) receptors [83]. Saikosaponin a, an active component of bupleuri radix, increased the duration of non-rapid eye movement sleep and shortened sleep latency by decreasing neuronal activity in the lateral hypothalamus in the male C57BL/6j mouse model [84].

 HM is thus believed to improve sleep through the above diverse and multiple mechanisms.”

Round 2

Reviewer 3 Report

All of the points pointed out in the previous review seem to be well reflected.

I "accept" the current version.